# Upstream Statin Therapy and Long-Term Recurrence of Atrial Fibrillation after Cardioversion: A Propensity-Matched Analysis

**DOI:** 10.3390/jcm10040807

**Published:** 2021-02-17

**Authors:** Lukas Fiedler, Lára Hallsson, Maximilian Tscharre, Sabrina Oebel, Michael Pfeffer, Robert Schönbauer, Lyudmyla Tokarska, Laura Stix, Anton Haiden, Johannes Kraus, Hermann Blessberger, Uwe Siebert, Franz Xaver Roithinger

**Affiliations:** 1Department of Internal Medicine, Cardiology, Nephrology and Intensive Care Medicine, Hospital Wiener Neustadt, 2700 Wiener Neustadt, Austria; maximilian.Tscharre@wienerneustadt.lknoe.at (M.T.); michael.Pfeffer@wienerneustadt.lknoe.at (M.P.); Lyudmyla.Tokarska@wienerneustadt.lknoe.at (L.T.); laura.stix@meduniwien.ac.at (L.S.); Anton.Haiden@wienerneustadt.lknoe.at (A.H.); FranzXaver.Roithinger@wienerneustadt.lknoe.at (F.X.R.); 2Institute of Public Health, Medical Decision Making and Health Technology Assessment, Department of Public Health, Health Services Research and Health Technology Assessment, UMIT—University for Health Sciences, Medical Informatics and Technology, 6060 Hall in Tirol, Austria; lara.hallsson@umit.at (L.H.); uwe.siebert@umit.at (U.S.); 3Department of Cardiology, Clinic of Internal Medicine II, Paracelsus Medical University of Salzburg, 5020 Salzburg, Austria; j.kraus@salk.at; 4Department of Cardiac Electrophysiology, Helios Heart Center Leipzig, University of Leipzig, 04289 Leipzig, Germany; s.r.oebel@gmx.at; 5Division of Cardiology, Department of Internal Medicine II, Medical University of Vienna, 1090 Vienna, Austria; robert.schoenbauer@meduniwien.ac.at; 6Department of Cardiology, Kepler University Hospital, 4040 Linz, Austria; hermann.blessberger@kepleruniklinikum.at; 7Johannes Kepler University Linz, Medical Faculty, 4040 Linz, Austria; 8Center for Health Decision Science and Departments of Epidemiology and Health Policy & Management, Harvard T.H. Chan School of Public Health, Boston, MA 02115, USA; 9Program on Cardiovascular Research, Institute for Technology Assessment and Department of Radiology and Department of Radiology, Massachusetts General Hospital, Harvard Medical School, Boston, MA 02115, USA

**Keywords:** statin, atrial fibrillation, cardioversion, recurrence

## Abstract

The relationship of statin therapy with recurrence of atrial fibrillation (AF) after cardioversion (CV) has been evaluated by several investigations, which provided conflicting results and particularly long-term data is scarce. We sought to examine whether upstream statin therapy is associated with long-term recurrence of AF after CV. This was a single-center registry study including consecutive AF patients (*n* = 454) undergoing CV. Cox regression models were performed to estimate AF recurrence comparing patients with and without statins. In addition, we performed a propensity score matched analysis with a 1:1 ratio. Statins were prescribed to 183 (40.3%) patients. After a median follow-up period of 373 (207–805) days, recurrence of AF was present in 150 (33.0%) patients. Patients receiving statins had a significantly lower rate of AF recurrence (log-rank *p* < 0.001). In univariate analysis, statin therapy was associated with a significantly reduced rate of AF recurrence (HR 0.333 (95% CI 0.225–0.493), *p* = 0.001), which remained significant after adjustment (HR 0.238 (95% CI 0.151–0.375), *p* < 0.001). After propensity score matching treatment with statins resulted in an absolute risk reduction of 27.5% for recurrent AF (21 (18.1%) vs. 53 (45.7%); *p* < 0.001). Statin therapy was associated with a reduced risk of long-term AF recurrence after successful cardioversion.

## 1. Introduction

Atrial fibrillation (AF) is the most prevalent arrhythmia in adults and according to epidemiological data the prevalence increases steadily in patients over 60 years of age [1]. Older age, hypertension, diabetes, atherosclerosis and heart failure are risk factors for the development and progression of AF, amongst others [2,3]. AF is associated with significant morbidity and mortality, and owing to the extending longevity of the general population, AF poses an increasing impact on health-care systems globally [3].

Restoration and maintenance of sinus rhythm (SR), known as rhythm control, resembles an integral part of the management of AF [3]. Electrical cardioversion is commonly used to restore SR, but also early pharmacological cardioversion or the combination of both strategies have been demonstrated to be effective [3]. Furthermore, in the recent EAST-AFNET 4 trial (Early Rhythm-Control Therapy in Patients with Atrial Fibrillation) early initiation of chronic rhythm control therapy was associated with a reduction of adverse cardiovascular outcomes compared to rate control [4]. However, despite the use of antiarrhythmic agents, a considerable proportion of patients relapse to AF and suffer from disease progression [5]. Several clinical and anatomical factors are associated with recurrence of AF, that is, higher age, atrial dilatation and longer duration of AF, and treatment of modifiable risk factors might facilitate maintenance of SR [2,6].

The role of inflammation in the development and progression of AF is well established [7]. Three-hydroxy-3-methylglutaryl coenzyme A (HMG-CoA) reductase inhibitors (Statins) effectively lower lipid levels. They are currently used as first-line therapy in primary and secondary prevention of patients suffering from hyperlipidemia and atherosclerotic disease, but also exhibit potent anti-oxidant and anti-inflammatory effects independent of their lipid lowering properties [8,9]. Statins increase nitric oxide synthesis, inhibit inflammatory cytokines, improve endothelial function, and can restore disturbed autonomic function [10,11,12,13]. Through these properties, statins are thought to contribute to the control of atrial fibrillation [14,15]. In experimental studies, statins prevented cardiac remodeling, and thus, might reduce the incidence of AF [16,17,18].

Although the relationship between statin therapy with AF recurrence has been evaluated by several observational studies and randomized trials, their results are conflicting and particularly long-term data are scarce [19,20,21]. Therefore, in this study, we examine whether upstream statin therapy is associated with reduced long-term recurrence of AF after successful cardioversion in patients with persistent and paroxysmal AF.

## 2. Materials and Methods

Patient population: We used a single-center registry including consecutive patients between 2012 and 2015 presenting with AF undergoing electrical or pharmacological cardioversion (CV). AF was defined as the absence of P waves, the presence of continuous and tachycardic atrial electrical activity and absolute irregular RR intervals on a resting 12-lead electrocardiogram (ECG). CV was performed according to current guidelines at that time, and the concomitant treatment and management of the patient was at the discretion of the attending cardiologist [22,23]. Patients with symptomatic paroxysmal AF with a duration of < 48 h underwent immediate CV. Patients with paroxysmal AF with an episode of AF > 48 h duration or with persistent AF (defined as AF > 7days) received oral anticoagulation over a period of four weeks before undergoing CV or received exclusion of intra-atrial thrombi using trans-esophageal echocardiography. Successful CV was defined as restoration of SR confirmed by a 12-lead ECG and maintenance of SR at the time of hospital discharge. Statin therapy was defined as ongoing intake of statins for at least three months prior to CV. To our knowledge, non-statin patients did not receive any statin therapy prior to CV.

Patients with postoperative AF were not eligible for this study.

The study was approved by the Medical Ethical Committee of Lower Austria (GS4-EK-4/164-2012) and informed consent was obtained from all patients.

Endpoint of interest: As endpoints of interest, we investigated the impact of statin therapy on the recurrence of AF over the course of three years. Recurrence of AF was defined as readmission for symptomatic AF or as the detection of asymptomatic AF episodes detected by ECG during opportunistic screening (either 12-lead-ECG or Holter ECG). Data were censored at time of adverse event or at end of follow-up. Outcome data were obtained by outpatient visits, using the regional hospital database system. or by telephone interviews.

Statistical analysis: Continuous variables are expressed as median (interquartile range [IQR]). Categorical variables are expressed as absolute numbers and percentages. Continuous variables were compared by Mann-Whitney-U-test or Fisher’s exact-test, as appropriate. χ2-tests were performed for categorical variables.

Survival curves were calculated using the Kaplan-Meier method and compared using the log-rank test. Cox proportional hazard models were performed to estimate adjusted hazard ratios with 95% confidence intervals (95%CI). Variable selection for the multivariable Cox proportional hazard models was applied according to the augmented backward elimination algorithm [24]. As a first step, all potential prognostic variables were included into a step-wise backward elimination model using a likelihood-ratio test with a significance level of α > 0.2 for exclusion. In a second step, all primarily excluded variables were re-entered separately and kept in the model in case of a change-in-estimate of > 5% in order to identify relevant confounders. Following variables were included into the primary model: Statin therapy, age, sex, hypertension, hyperlipidaemia, diabetes, heart failure, presence of atherosclerotic disease, prior ablation, and concomitant pharmacological therapy. Partial residuals were used to assess the proportional hazard assumption. All variables, included in the final Cox regression model, fulfilled the proportional hazard assumption.

In addition, we performed a propensity score matched analysis using the aforementioned variables. Propensity score matching was performed using a 1:1 ratio using nearest neighbour matching with a caliper width of 0.2. The balance of risk factors was judged by standardized differences. No covariate exhibited a large imbalance.

All statistical tests were 2-tailed, and a *p*-value < 0.05 was considered statistically significant. All statistical analyses and figures were generated with SPSS V21.0 (IBM®SPSS®Statistics V21.0, IBM Deutschland GmbH, 71137 Ehningen, Baden-Württemberg, Germany) and R 3.6.3 (Copyright © 1989, 1991 Free Software Foundation, Inc., 59 Temple Place, Suite 330, Boston, MA 02111-1307 USA).

## 3. Results

In total, 454 patients were eligible for the present analysis. Of all patients, 183 (40.3%) patients received statin therapy at time of cardioversion. Median age was 65 (IQR 57–72) years, and 146 (32.2%) were female. Paroxysmal AF was present in 198 (43.6%) patients and persistent AF in 256 (56.4%) patients. Electrical CV was performed in 438 (96.5%) patients and pharmacological CV was performed in 16 (3.5%) patients. Statins were prescribed in 183 (40.3%) patients. Clinical and procedural characteristics of the overall study population and stratified for patients with and without statins are shown in Table 1. Types of statins, received by patients, are depicted in Table 2.

There were substantial differences in baseline parameters between both groups. Patients with statins were older (*p* < 0.001), had a higher CHA2DS2-VASc score (*p* < 0.001), and a higher prevalence of arterial hypertension (*p* < 0.001), diabetes (*p* < 0.001), coronary artery disease (*p* < 0.001), heart failure (*p* = 0.046) and hyperlipidaemia (*p* < 0.001).

### Clinical Outcomes

After a median follow-up period of 373 (207–805) days recurrence of AF was present in 150 (33.0%) patients (Table 1). Patients receiving statins had a significantly lower rate of AF recurrence when compared to patients not receiving statins (log rank *p* < 0.001; crude HR 0.333 [95% CI 0.225–0.493], *p* = 0.001), as shown in Figure 1. After adjustment for confounders, statin therapy remained significantly associated with a reduced rate of recurrence of AF (adjusted HR 0.238 [95% CI 0.151–0.375], *p* < 0.001). Also, when stratified for sex, statin therapy was associated with a reduction of recur-rent AF after multivariable adjustment (male patients: HR 0.262 [95%CI 0.151–0.454], *p* < 0.001; female patients: HR 0.178 [95%CI 0.078–0.404], *p* < 0.001). Also, in another sub-analysis, both high-intensity statins (atorvastatin or rosuvastatin) and moderate-intensity statins (all other statins) were both associated with reduced AF recurrence after multivariable adjustment (high-intensity statins: HR 0.203 [95%CI 0.080–0.510], *p* = 0.001; moderate-intensity statins: HR 0.245 [95%CI 0.150–0.401], *p* < 0.001). Due to the substantial differences in baseline parameters for the two study groups, a propensity score matched analysis was performed generating 116 matched pairs. After propensity score matching for confounders, statin therapy resulted in an absolute risk reduction of 27.5% for recurrent AF (21 [18.1%] vs. 53 [45.7%]; *p* < 0.001; Table 1 and Figure 2). Predictors of recurrent AF are presented in Table 3.

## 4. Discussion

According to the results of our study upstream concomitant statin therapy was significantly associated with a reduction in AF recurrence after successful cardioversion during a median follow-up period of 373 days. Using propensity score matched analysis, statin therapy resulted in an absolute risk reduction of 27.5% for recurrent AF.

AF is caused by micro re-entrant circuits originating from the pulmonary veins and its development is associated with advanced age and multiple cardiovascular risk factors [2,3]. With increasing duration, AF results in structural and electrical remodeling of the atria, which further rises susceptibility for the initiation and propagation of arrhythmia [25]. The induction of AF is critically linked with inflammatory processes, including oxidative stress, endothelial dysfunction and vascular inflammation [26,27,28].

Statins competitively inhibit HMG-CoA reductase, which resembles the rate-limiting step in hepatic cholesterol biosynthesis [8,9]. As hypercholesterolemia has been associated with up-regulation of KACh cardiac currents in atrial cardiomyocytes, statins might expose beneficial effects on AF [29]. Apart from lipid lowering effects, statins have also been associated with additional pleiotropic effects resulting in reduced vascular inflammation, decreased oxidative stress and improved endothelial function [9,30]. In several animal AF models, treatment with statins was associated with reduced atrial dilatation and fibrosis, both resembling major risk factors for the development of AF [16,17,18]. These beneficial effects of statins were attributed to the inhibition of nicotinamide adenine dinucleotide phosphate (NADPH), oxidase, angiotensin II, myeloperoxidase (MPO), several metalloproteinases (MMPs) and its inhibitors (TIMPs; tissue inhibitors of metalloproteinase) by statins, known effectors of oxidative stress and extracellular matrix metabolism [16,31,32].

Based on these experimental data and our results statin treatment potentially represents an attractive candidate for upstream therapy before cardioversion. Nonetheless, several observational studies and randomized trials investigated the impact of statins on early recurrence of AF with conflicting results [19,20]. However, the majority of the neutral studies were limited by small patient numbers and short follow-up periods [19,20]. Also, in most randomized trials statin therapy was initiated shortly before cardioversion (two days to three weeks) and only maintained for a brief period (three weeks up to 3 months) [33,34,35,36]. According to prior data anti-inflammatory effects of statins measured by CRP-levels start to change significantly only after 4 months [37]. In this regard the follow-up and treatment duration of the existing randomized controlled trials (RCTs) demonstrating neutral effects of statin therapy on AF recurrence might have been too short to document a significant effect [19,20]. These factors might contribute to the conflicting reports of statins and SR maintenance, as data from a population-based study demonstrated a time-dependent effect of statins and the risk of AF or atrial flutter, in which beneficial effects were detectible only in patients with long-term exposure to statins [38]. Likewise, in a small observational study beneficial effects of statin use on SR maintenance were only apparent after six months of treatment [39].

AF is a chronic disease requiring rigorous and long-term management of risk factors, in order to inhibit disease progression and preserve SR [6]. Likewise, our data indicate a relevant impact of statin therapy on long-term recurrence of AF. Further large clinical trials are needed to elucidate the role of chronic statin therapy in long-term SR maintenance in patients after cardioversion for AF. Interestingly, in our cohort hyperlipidemia was associated with recurrent AF. In line with our finding, Balse et al. demonstrated modulating effects of cholesterol on potassium channels in atrial myocytes [40]. Therefore, apart from their pleotropic and anti-inflammatory effects, statins might also prevent recurrent AF by lowering cholesterol levels. This assumption was not confirmed in our sub-analysis comparing high-intensity with moderate-intensity statins, although there was a trend towards better efficacy for high-intensity statins. However, our study population was not powered for a definite conclusion.

Our data must be regarded as merely hypothesis-generating only and further investigations are needed, as this study cannot prove causality nor exclude the possibility of unmeasured confounding. The present investigation should be interpreted with the following limitations in mind. The results were derived from a single-center registry including patients undergoing cardioversion for AF. No LDL-C levels were collected to monitor statin therapy. There was no documentation of any echocardiography parameters and adherence to statin therapy. Furthermore, there was no structured protocol for detection of silent or asymptomatic AF.

Finally, the results of our registry study should be seen as a motivation for further methodological analyses explaining conflicting results from randomized clinical trials and observational studies. Causal inference methods of comparative effectiveness analysis such as g-methods [41] and the target trial approach [42] should be used to emulate clinical trials, based on real world evidence of observational studies, affected by time-varying confounding.

## 5. Conclusions

Statin therapy was associated with reduced risk of long-term AF recurrence after successful cardioversion. Further large clinical trials are needed to elucidate the causal role of statin therapy in SR maintenance in patients after cardioversion for AF.

## Figures and Tables

**Figure 1 jcm-10-00807-f001:**
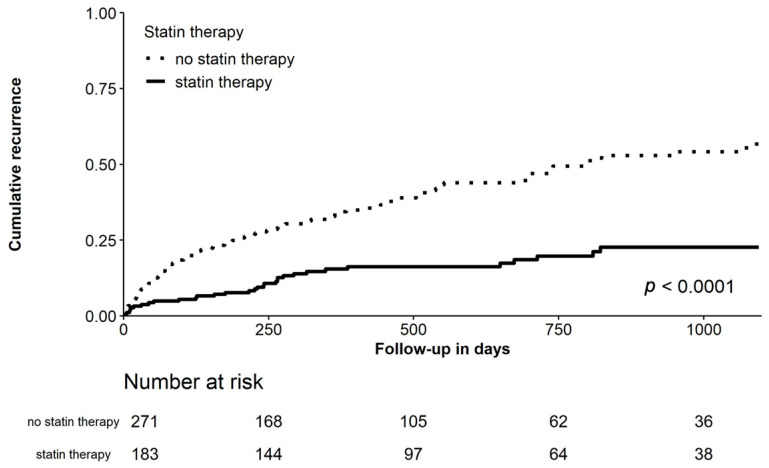
Cumulative incidence curve for recurrence of atrial fibrillation after CV comparing all patients with and without statins, with the log-rank test.

**Figure 2 jcm-10-00807-f002:**
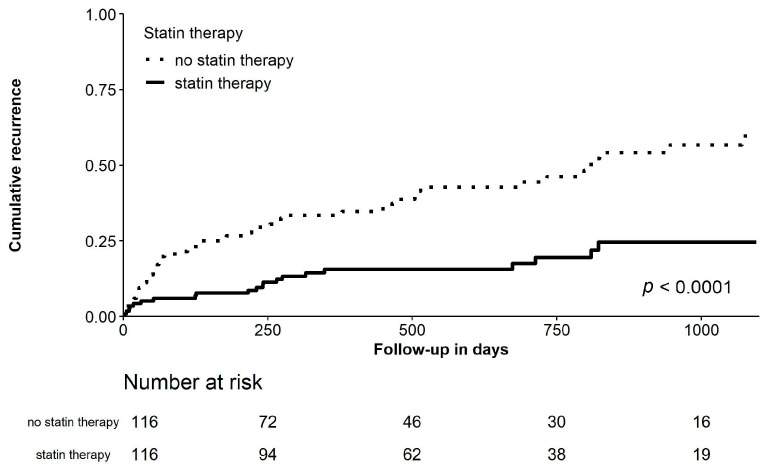
Cumulative incidence curve for recurrence of atrial fibrillation comparing patients with and without statins using propensity-matched pairs, with log-rank test.

**Table 1 jcm-10-00807-t001:** Baseline characteristics of study population.

		Before Matching			After Matching		
	All Patients	No Statin	Statin	*p*-Value	No Statin	Statin	*p*-Value
	*n* = 454	*n* = 271	*n* = 183		*n* = 116	*n* = 116	
Recurrence of AF, No. (%)	150 (33.0%)	118 (43.5%)	32 (17.5%)	<0.001	53 (45.7%)	21 (18.1%)	<0.001
Age, years	65 (57–72)	64 (54–71)	67 (61–72)	0.001	67 (61–72)	66 (60–71)	0.427
Female patients, No. (%)	146 (32.2%)	87 (32.1%)	59 (32.2%)	1.000	43 (37.1%)	38 (32.8%)	0.582
Clinical presentation, No. (%)							
persistent AF	256 (56.4%)	155 (57.2%)	101 (55.2%)	0.744	66 (56.9%)	61 (52.6%)	0.598
paroxysmal AF	198 (43.6%)	116 (42.8%)	82 (44.8%)		50 (43.1%)	55 (47.4%)	
Type of cardioversion, No. (%)							
Electrical	438 (96.5%)	257 (94.8%)	181 (98.9%)	0.040	109 (94.0%)	115 (99.1%)	0.073
Pharmacological	16 (3.5%)	14 (5.2%)	2 (1.1%)		7 (6.0%)	1 (0.9%)	
CHA_2_DS_2_-VASc	2 (1–2)	2 (1–2)	2 (2–3)	<0.001	2 (2–3)	2 (1–2)	0.477
Prior ablation, No. (%)	78 (17.2%)	47 (17.3%)	31 (16.9%)	1.000	19 (16.4%)	19 (16.4%)	1.000
Arterial hypertension, No. (%)	313 (68.9%)	165 (60.9%)	148 (80.9%)	<0.001	92 (79.3%)	88 (75.9%)	0.637
Diabetes mellitus, No. (%):	66 (14.5%)	25 (9.23%)	41 (22.4%)	<0.001	20 (17.2%)	24 (20.7%)	0.503
Coronary artery disease, No. (%)	75 (16.5%)	26 (9.6%)	49 (26.8%)	<0.001	22 (19.0%)	22 (19.0%)	1.000
Heart failure, No. (%)	83 (18.3%)	41 (15.1%)	42 (23.0%)	0.046	27 (23.3%)	22 (19.0%)	0.520
Hyperlipidemia, No. (%)	111 (24.4%)	23 (8.49%)	88 (48.1%)	<0.001	23 (19.8%)	26 (22.4%)	0.748
Beta blocker, No. (%)	304 (67.0%)	178 (65.7%)	126 (68.9%)	0.547	78 (67.2%)	77 (66.4%)	1.000
ACE inhibitor, No. (%)	175 (38.5%)	96 (35.4%)	79 (43.2%)	0.118	63 (54.3%)	51 (44.0%)	0.149
ARB, No.(%)	101 (22.2%)	53 (19.6%)	48 (26.2%)	0.118	25 (21.6%)	26 (22.4%)	1.000
MRA, No.(%)	38 (8.41%)	18 (6.69%)	20 (10.9%)	0.155	11 (9.48%)	9 (7.76%)	0.815
Anti-arrhythmic therapy, No. (%)	303 (66.7%)	176 (64.9%)	127 (69.4%)	0.375	81 (69.8%)	80 (69.0%)	1.000
Amiodarone	220 (48.5%)	123 (45.4%)	97 (53.0%)	0.111	54 (46.6%)	59 (50.9%)	0.599
Sotalol	27 (5.9%)	18 (6.6%)	9 (4.9%)	0.446	8 (6.9%)	7 (6.0%)	1.000
Dronedarone	23 (5.1%)	14 (5.2%)	9 (4.9%)	0.906	5 (4.3%)	6 (5.2%)	1.000
Propafenone	16 (3.5%)	10 (3.7%)	6 (3.3%)	0.816	7 (6.0%)	6 (5.2%)	1.000
Flecainide	17 (3.7%)	11 (4.1%)	6 (3.3%)	0.667	7 (6.0%)	2 (1.7%)	0.171
Digitalis, No. (%)	38 (8.4%)	15 (5.5%)	23 (12.6%)	0.013	13 (11.2%)	15 (12.9%)	0.840

Data are presented as median (IQR) or numbers (%). ACE: angiotensin converting enzyme; AF: atrial fibrillation; ARB: angiotensin receptor blocker; MRA: mineralocorticoid receptor antagonist; No.: numbers.

**Table 2 jcm-10-00807-t002:** Type of statin prescribed in this study.

	Patients with Statins
	*n* = 183
Fluvastatin, No. (%)	17 (9.3)
Pravastatin, No. (%)	18 (9.8)
Simvastatin, No. (%)	112 (61.2)
Atorvastatin, No. (%)	31 (16.9)
Rosuvastatin, No. (%)	5 (2.7)

Data are presented as numbers (%).

**Table 3 jcm-10-00807-t003:** Predictors of recurrent atrial fibrillation for the propensity-matched cohort using the augmented backward elimination algorithm.

Final Model (Continuous)				
	HR	95% CI		*p*-Value
Statin therapy	0.313	0.188	0.521	<0.001
Hyperlipidemia	3.018	1.860	4.895	<0.001
Anti-arrhythmic therapy	1.443	0.858	2.427	0.166

Legend: HR: hazard ratio; CI: confidence interval.

## Data Availability

The data presented in this study are available on request from the corresponding author.

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
