# Peer review of "Upstream Statin Therapy and Long-Term Recurrence of Atrial Fibrillation after Cardioversion: A Propensity-Matched Analysis"

_jcm, 2021, doi:10.3390/jcm10040807_

Round 1

Reviewer 1 Report

It is with interest that I have reviewed the manuscript entitled: upstream statin therapy and long-term recurrence of atrial fibrillation after cardioversion: A propensity-matched analysis. [jcm-1072507]. The study investigates the relationship between statin therapy and long-term AF recurrence after cardioversion. The paper is well written and covers an adequate path from the hypothesis to the results, discussion and conclusion. Here are some comments:

  1. Introduction: Could the author provide precisions around the incidence of AF reported to be increased around the last decades of life? Is there specifics around the age above which AF incidence is known to increase?
  2. Line 65: Following the statement that statins increase nitric oxide synthesis, inhibit inflammatory cytokines, improve endothelial function, and restore autonomic function, it would adequate for the authors to add references for each of these mechanistic properties from which statins have been studies whether across in-vitro/in-vivo or clinical studies.
  3. Did the author test whether the association between statins and long-term recurrence of AF could be more or less pronounced when adjusting to high-intensity vs non-high intensity statins?
  4. Table 2. Do we know the treatment regimen (dosage) and period of treatment for statins therapies? As authors stated: statin therapy was defined as ongoing intake for at least three month. Could the authors speculate if long-term lipid-lowering treatments would have a potential effect on AF recurrence?
  5. What can we speculate on lipid parameters in regards to long-term AF recurrence. In other words: Did the authors collect data in respect to LDL-C levels at baseline (between statin and non-statin) or across follow-up analysis? Statin compliance and adherence is a challenging aspect and LDL-C levels across long-term follow-up could somehow reflect statin efficacy.
  6. Line 163: As inflammation is an important process in AF and statin therapy seems to reduce recurrence of AF after CV due to anti-inflammatory effect among others; I would suggest for the authors to add more references in regards to the critical link of AF with inflammatory process, as mentioned, in regards to ox. stress, end. dys. and vascular inflammation.
  7. It would be interesting for the audience to consider mentioning if any sex differences (male vs female) are noted in respect to AF recurrence between statin and non-statin.
  8. Line 61: It is less appropriate to start a sentence with a number. I suggest for the author to replace it with: Three-hydroxy-3-mthylglutaryl coenzyme A (HMG-CoA) reductase.

Author Response

RESPONSE LETTER

Dear editor,

We would like to submit a revised version of our manuscript entitled “Upstream statin therapy and long-term recurrence of atrial fibrillation after cardioversion: A propensity-matched analysis” for publication in Journal of Clinical Medicine. We are grateful for the opportunity to answer the questions, criticisms and comments raised by the referees. The manuscript has been amended accordingly. We believe that these changes have resulted in a greatly improved manuscript which we hope is now suitable for publication in Journal of Clinical Medicine.

Best regards

Lukas Fiedler, MD

Point-by-point response to reviewer: 1:

Thank you for carefully reading our manuscript and your important suggestions which we have followed.

It is with interest that I have reviewed the manuscript entitled: upstream statin therapy and long-term recurrence of atrial fibrillation after cardioversion: A propensity-matched analysis. [jcm-1072507]. The study investigates the relationship between statin therapy and long-term AF recurrence after cardioversion. The paper is well written and covers an adequate path from the hypothesis to the results, discussion and conclusion. Here are some comments:

  1. Introduction: Could the author provide precisions around the incidence of AF reported to be increased around the last decades of life? Is there specifics around the age above which AF incidence is known to increase?

Response: According to epidemiological data the prevalence of AF increases steadily in patients over 60 years of age (EHJ, Apr. 2006, 27, 8: 949–953). We have added a statement in the introduction section accordingly (please see page 1, line 43).

  1. Line 65: Following the statement that statins increase nitric oxide synthesis, inhibit inflammatory cytokines, improve endothelial function, and restore autonomic function, it would adequate for the authors to add references for each of these mechanistic properties from which statins have been studies whether across in-vitro/in-vivo or clinical studies.

Response: We added specific references as you suggested for each mechanistic property. (please see page2, line 111).

  1. Did the author test whether the association between statins and long-term recurrence of AF could be more or less pronounced when adjusting to high-intensity vs non-high intensity statins?

Response: In a sub-analysis, both types of statins were similarly associated with a reduced rate of AF recurrence. We have adapted the results section accordingly (please see page 5, line 160).

  1. Table 2. Do we know the treatment regimen (dosage) and period of treatment for statins therapies? As authors stated: statin therapy was defined as ongoing intake for at least three months. Could the authors speculate if long-term lipid-lowering treatments would have a potential effect on AF recurrence?

Response: Unfortunately, treatment duration was not assessed more specifically. We have revised the limitation section accordingly.

According to de Lemos et al. (JAMA 2004, 292, 11:1307) anti-inflammatory effects estimated by C-reactive protein were only seen after four months of treatment. We therefore speculate, that only long-term therapy might have a relevant impact on AF recurrence. We have added a statement in the discussion section (page 6, line 208)

  1. What can we speculate on lipid parameters in regards to long-term AF recurrence. In other words: Did the authors collect data in respect to LDL-C levels at baseline (between statin and non-statin) or across follow-up analysis? Statin compliance and adherence is a challenging aspect and LDL-C levels across long-term follow-up could somehow reflect statin efficacy.

Response: Unfortunately, we did not collect data on statin adherence or LDL-C levels. We have revised the limitations section accordingly (page 7, line 227)

  1. Line 163: As inflammation is an important process in AF and statin therapy seems to reduce recurrence of AF after CV due to anti-inflammatory effect among others; I would suggest for the authors to add more references in regards to the critical link of AF with inflammatory process, as mentioned, in regards to ox. stress, end. dys. and vascular inflammation.

Response: We added specific references as you suggested. (please see page 6, line 186).

  1. It would be interesting for the audience to consider mentioning if any sex differences (male vs female) are noted in respect to AF recurrence between statin and non-statin.

Response: Statin therapy was associated with a reduction of AF recurrence regardless of sex, even after multivariable adjustment. We have revised the results section accordingly (please see page 5, line 157).

  1. Line 61: It is less appropriate to start a sentence with a number. I suggest for the author to replace it with: Three-hydroxy-3-mthylglutaryl coenzyme A (HMG-CoA) reductase.

Response: We have adopted the manuscript accordingly (please see page 2, line 68).

Point-by-point response to reviewer 2:

Thank you for carefully reading our manuscript and your important suggestions which we have followed.

In this single centre retrospective study the authors investigated the impact of statin therapy on atrial fibrillation recurrence rates in patients undergoing successful cardioversion. Among 454 consecutive patients with atrial fibrillation, the authors selected two groups of 116 1:1 propensity matched patients. The primary end-point was atrial fibrillation recurrence, assessed through outpatient visits or telephone interview. After a median follow-up of 373 days, the authors observed that statin therapy is associated with 27.5% absolute risk reduction of atrial fibrillation recurrence (p<0.001). The authors concluded statin therapy is associated with reduced risk of long-term atrial fibrillation recurrence after successful cardioversion.

1) Material and methods section, page 2, line 88-89, “Statin therapy was defined as ongoing intake for at least three months”: Statin therapy definition results unclear: were all patients in statin group taking statins for more than 3 months before the cardioversion? Were all non-statin patients stain naïve or patients taking statins for less than three months before the cardioversion event included? Patients prescribed with statins at the time of the cardioversion were included in the non-statin group? Please clarify.

Response: All patients were on statins for more than 3 months. Patients in the non-statin group did not receive a statin.

We added an additional paragraph to better specify the two groups statin- and no-statin therapy: (please see page 2, line 132)

  1. Material and methods section, page 2, line 93-96, “Endpoint of interest … or by telephone interviews.”: The primary end-point definition is unclear. Were all patients screened the same way for atrial fibrillation recurrence? Were fixed outpatient visits planned or only recurrence of symptomatic atrial fibrillation episodes were detected? Atrial fibrillation detection through opportunistic ECGs or Holter registration allowed? Please clarify this point in order to avoid possible biases in the primary endpoint assessment.

Response: Recurrence of AF was defined as readmission for symptomatic AF or as the detection of asymptomatic AF episodes documented by ECG during opportunistic screening (either 12-lead or Holter-ECG). All patients who were not referred for recurrent AF to our department were both screened using the regional hospital database system and using telephone interviews. We have adopted the methods section accordingly (please see page 3, line 103).

  1. Results section, page 3, lines 125-126, “Statins were prescribed in 183 (40.3%) of patients.”: No data on statin therapy compliance is provided. How many of the statin prescribed patients were still on statin at follow-up? There was any crossover among groups during follow-up and how were those patients managed? If those data are available, please specify.

Response: Unfortunately, we did not collect data on statin adherence. We have revised the limitations section accordingly (page 7, line 227)

  1. Anti-arrhythmic therapy should be better described. Furthermore, no information about time on AF for persistent AF has been provided, and no information about atrial sizes, valvular involvement, systolic and diastolic function by echocardiography has been reported and investigated. These are important predictors of AF recurrences.

Response: We have adopted Table 1 accordingly. Unfortunately, we have do not have echocardiography parameters for our study population. We have added a statement in the limitations section (please see page 7, line 225)

  1. Discussion section, page 6, line 169, “pleotropic”:please correct with “pleiotropic”. Please check also the manuscript for typos.

Response: We have revised the manuscript accordingly and have screened the manuscript for additional typographical errors.

  1. A table for predictors of AF recurrence should be provided.

Response: We now demonstrate the predictors of AF recurrence in Table 3.

  1. Speculative analysis about statin type and dosage can be of interest (but maybe underpowered).

Response: In a sub-analysis, both types of statins (high-intensity vs. moderate-intensity statins) were similarly associated with a reduced rate of AF recurrence. We have adapted the results section accordingly (please see page 5, line 160).

The authors concluded statin therapy is associated with reduced risk of long term atrial fibrillation recurrence after successful cardioversion but this in only hypothesis-generating as acknowledged by the authors, due to the retrospective nature of the work and possible non-investigated confounding factors. The manuscript is well-written, length is adequate, and the language is enough clear. The title clearly resumes the intents of the research. Methods should be empowered with a clear definition of statin and non-statin therapy groups. Methods for atrial fibrillation recurrence detection should also be clarified, since a possible detection bias might have occurred (see major issues). The absence of investigation for silent AF is a major limit. Missing data should be provided if possible (echocardiography, anti-arrhythmic therapy, time on AF), since they may provide important predictor for AF recurrence. Results are brief and concise. Conclusions are clear.

  1. Novelty is not so high, since RCT an meta-analysis just exist about this issue. Authors should be better clarified differences and novelties from previous works, if present.

Response: Although novelty is not high, our data strengthen the premise that particularly long-term statin therapy might be associated with beneficial effects on AF recurrence. Only RCTs with a longer therapy at least could demonstrate a trend favouring statin therapy after CV.

Reviewer 2 Report

In this single centre retrospective study the authors investigated the impact of statin therapy on atrial fibrillation recurrence rates in patients undergoing successful cardioversion. Among 454 consecutive patients with atrial fibrillation, the authors selected two groups of 116 1:1 propensity matched patients. The primary end-point was atrial fibrillation recurrence, assessed through outpatient visits or telephone interview. After a median follow-up of 373 days, the authors observed that statin therapy is associated with 27.5% absolute risk reduction of atrial fibrillation recurrence (p<0.001). The authors concluded statin therapy is associated with reduced risk of long term atrial fibrillation recurrence after successful cardioversion.

MAJOR ISSUES:

Material and methods section, page 2, line 88-89, “Statin therapy was defined as ongoing intake for at least three months”: Statin therapy definition results unclear: were all patients in statin group taking statins for more than 3 months before the cardioversion? Were all non-statin patients stain naïve or patients taking statins for less than three months before the cardioversion event included? Patients prescribed with statins at the time of the cardioversion were included in the non-statin group? Please clarify.

Material and methods section, page 2, line 93-96, “Endpoint of interest … or by telephone interviews.”: The primary end-point definition is unclear. Were all patients screened the same way for atrial fibrillation recurrence? Were fixed outpatient visits planned or only recurrence of symptomatic atrial fibrillation episodes were detected? Atrial fibrillation detection through opportunistic ECGs or Holter registration allowed? Please clarify this point in order to avoid possible biases in the primary endpoint assessment.

Results section, page 3, lines 125-126, “Statins were prescribed in 183 (40.3%) of patients.”: No data on statin therapy compliance is provided. How many of the statin prescribed patients were still on statin at follow-up? There was any crossover among groups during follow-up and how were those patients managed? If those data are available please specify.

Anti-arrhythmic therapy should be better described. Furthermore, no information about time on AF for persistent AF has been provided, and no information about atrial sizes, valvular involvement, systolic and diastolic function by echocardiography has been reported and investigated. These are important predictors of AF recurrences.

MINOR ISSUES:

Discussion section, page 6, line 169, “pleotropic”: please correct with “pleiotropic”. Please check also the manuscript for typos.

A table for predictors of AF recurrence should be provided.

Speculative analysis about statin type and dosage can be of interest (but maybe underpowered).

FINAL COMMENTS

The authors concluded statin therapy is associated with reduced risk of long term atrial fibrillation recurrence after successful cardioversion but this in only hypothesis-generating as acknowledged by the authors, due to the retrospective nature of the work and possible non-investigated confounding factors. The manuscript is well-written, length is adequate and the language is enough clear. The title clearly resumes the intents of the research. Methods should be empowered with a clear definition of statin and non-statin therapy groups. Methods for atrial fibrillation recurrence detection should also be clarified, since a possible detection bias might have occurred (see major issues). The absence of investigation for silent AF is a major limit.

Missing data should be provided if possible (echocardiography, anti-arrhythmic therapy, time on AF), since they may provide important predictor for AF recurrence. 

Results are brief and concise. Conclusions are clear.

Novelty is not so high, since RCT an meta-analysis just exist about this issue. Authors should be better clarified differences and novelties from previous works, if present.

Author Response

(The authors gave the same response as above.)

Round 2

Reviewer 2 Report

The author's have correctly addressed all my previous comments. The quality of the paper has increased. Table 3 is surprising. Hyperlipidemia is a strong predictor of AF recurrence while statin use is protective. I guess hyperlipidemia definition comprises only those patients not on statin therapy. But, if so, the role of statins is at least in part dependent from lipid lowering. This is not confirmed by sub-analysis about statin intensity. Author's should be better discuss this finding.

Author Response

RESPONSE LETTER

Dear editor,

We would like to submit a revised version of our manuscript entitled “Upstream statin therapy and long-term recurrence of atrial fibrillation after cardioversion: A propensity-matched analysis” for publication in Journal of Clinical Medicine. We are grateful for the opportunity to answer the questions, criticisms and comments raised by the referees. The manuscript has been amended accordingly. We believe that these changes have resulted in a greatly improved manuscript which we hope is now suitable for publication in Journal of Clinical Medicine.

Best regards

Lukas Fiedler, MD

Point-by-point response to reviewer 2

Thank you for carefully reading our manuscript, your kind comments and important suggestions which we have tried to follow.

Point 1: The author's have correctly addressed all my previous comments. The quality of the paper has increased. Table 3 is surprising. Hyperlipidemia is a strong predictor of AF recurrence while statin use is protective. I guess hyperlipidemia definition comprises only those patients not on statin therapy. But, if so, the role of statins is at least in part dependent from lipid lowering. This is not confirmed by sub-analysis about statin intensity. Author's should be better discuss this finding. 

Response Point 1:  see Page 7, Line 218-225

Interestingly, in our cohort hyperlipidemia was associated with recurrent AF. In line with our finding, Balse et al. demonstrated modulating effects of cholesterol on potassium channels in atrial myocytes. (Proc. Natl. Acad. Sci. U. S. A. 2009, 106, 14681–14686, doi:10.1073/pnas.0902809106.) Therefore, apart from their pleotropic and anti-inflammatory effects, statins might also prevent recurrent AF by lowering cholesterol levels. This assumption was not confirmed in our sub-analysis comparing high-intensity with moderate-intensity statins, although there was a trend towards better efficacy for high-intensity statins. However, our study population was not powered for a definite conclusion.